# BrainSF: A Foundation Model for Whole Brain Functional Signals Forecasting

## Abstract

Foundational models hold significant potential for advancing brain function research, particularly with recent technological advancements enabling the capture of spatiotemporal dynamics of brain signals. However, existing methods are primarily limited to characterizing observed brain signals and cannot infer continuous future signals—an essential component for understanding the brain's causal structure and its role in various cognitive states. Current research leaves a substantial gap in forecasting whole-brain signal sequences. To address this, we propose a self-supervised model that embeds momentary whole-brain fMRI signals into vector representations and predicts continuous future signals. Our model is trained on a large-scale fMRI dataset, encompassing both resting-state and naturalistic stimuli conditions. Experimental results demonstrate that the model performs effectively in zero-shot forecasting of future whole-brain signals on unseen data and excels in downstream tasks such as task-based functional state decoding. To the best of our knowledge, this is the first approach to forecast and model whole-brain signals at such a large scale. The experimental results validate the feasibility of our method, offering new directions for theoretical research on brain signal time series and potential applications in diagnosing and treating brain disorders.

## 1 Introduction

Functional magnetic resonance imaging (fMRI) has revolutionized our ability to non-invasively study brain function by providing rich spatiotemporal data of whole-brain neural activity (Van Essen et al., 2013a). However, analyzing and modeling the complex dynamics of brain signals remains a significant challenge (Bzdok et al., 2020). Recent advances in machine learning, particularly the development of deep learning methods, offer promising new approaches for capturing intricate patterns and relationships within brain functional data (Thomas et al., 2022).

Deep learning models, which learn general-purpose representations through training on large-scale datasets, have achieved remarkable success across various domains of artificial intelligence (LeCun et al., 2015). In neuroscience, these models hold great potential for advancing our understanding of brain function by learning robust and generalizable representations of neural activity patterns (Schirrmeister et al., 2017). Recent studies have introduced techniques that employ deep learning to capture the spatiotemporal dynamics of brain signals (Kong et al., 2022). However, existing methods (Ortega Caro et al., 2023; Thomas et al., 2022; Yang et al., 2024) are limited to describing observed brain signals and cannot infer continuous future signals—a capability crucial for understanding the brain's causal structure and its roles in different cognitive states.

This limitation highlights the need for more sophisticated approaches, especially in the domain of brain signal sequence prediction (Shine & Breakspear, 2023). Accurately predicting future signals from past signals can provide valuable insights into the temporal evolution of neural processes, cognitive dynamics, and causal relationships between brain regions. Such predictive capabilities can significantly enhance our ability to study complex cognitive phenomena, diagnose neurological disorders, and develop brain-computer interfaces.

In this paper, we present a novel self-supervised deep learning model designed for whole brain functional signal forecasting, called BrainSF. Our model embeds instantaneous whole brain signals into vector representations and predicts continuous future signals. BrainSF is trained on large-scale fMRI datasets encompassing resting-state and naturalistic stimulus conditions (Van Essen et al.,

2013a; Nastase et al., 2021b), enabling it to learn generalizable representations of brain dynamics across diverse cognitive states. We focus on resting-state and naturalistic stimulus data for two primary reasons: first, these data types are relatively accessible, allowing for the collection of large samples to support large-scale model training (Turner & Calhoun, 2023); second, compared to traditional task-based fMRI, brain activity under resting-state and naturalistic conditions encompasses a broader and richer set of functional networks, providing the model with a more comprehensive representation of brain dynamics (Simony et al., 2016). This approach enables our model to capture more generalized patterns of brain function, thereby demonstrating superior performance across various downstream tasks.

Our main contributions are summarized as follows:

- We propose a novel foundational model architecture specifically tailored for brain functional signal prediction, capable of handling input and output sequences of varying lengths, and trained on large-scale, diverse fMRI datasets.
- We demonstrate the model's strong zero-shot signal prediction performance on unseen data, validating the generalization capability of the learned representations and the model's effectiveness in downstream tasks such as task-state decoding.
- We conduct an in-depth analysis of the representations learned by the model, revealing how it captures key features of brain signal dynamics across continuous time dimensions. This opens new avenues for theoretical studies of brain signal time series and potential applications in diagnosing and treating brain disorders.

To the best of our knowledge, this is the first approach capable of predicting and modeling brain signals at such a scale. Our experimental results validate the feasibility of this method, offering promising directions for theoretical research on brain signal time series and potential applications in diagnosing and treating brain disorders. In the subsequent sections, we detail our methodology, experimental setup, and results, and discuss the broader implications of this work for the fields of neuroscience and artificial intelligence. We will make the code and model publicly available upon acceptance.

## 2 RELATED WORK

### 2.1 TIME SERIES FORECASTING

Recent progress in deep learning has greatly advanced time series forecasting. Although recurrent neural networks (RNNs) such as long short-term memory (LSTM) networks (Hochreiter & Schmidhuber, 1997) and gated recurrent units (GRUs) (Cho et al., 2014) are effective at capturing temporal patterns, they face limitations with handling very long sequences. To address this, researchers have turned to alternative models like temporal convolutional networks (TCNs) (Bai et al., 2018) and Transformers (Vaswani et al., 2017; Zhou et al., 2021; Wu et al., 2021). Moreover, strategies such as attention mechanisms (Qin et al., 2017), multi-task learning (Rodrigues & Pereira, 2018), and transfer learning (Laptev et al., 2018) have improved forecasting accuracy, helping to tackle issues like missing data (Che et al., 2018), anomaly detection (Hundman et al., 2018), and uncertainty quantification (Zhu & Laptev, 2017). Graph neural networks (GNNs) have also shown potential, as they can model both temporal and spatial dependencies (Bai et al., 2022; Shang et al., 2023).

### 2.2 BRAIN FUNCTIONAL SIGNAL REPRESENTATION

Recent advancements in deep learning have significantly improved brain functional signal representation. Techniques such as convolutional autoencoders (CAEs) (Li et al., 2018) effectively learn compact and informative representations of fMRI data by capturing spatial patterns and temporal dynamics. To model the intrinsic structure of brain networks, graph convolutional networks (GCNs) and graph autoencoders (Arslan et al., 2018) preserve topological structures, with spatio-temporal GCNs (Gadgil et al., 2021) incorporating temporal dynamics for time-varying signals. Attention mechanisms further enhance representational power: attention-based LSTMs (Mahmud et al., 2020) focus on relevant spatial and temporal features, and graph attention networks (GATs) (Jiang et al., 2022) assign different weights to neighboring nodes for more expressive and interpretable representations.

## 2.3 FOUNDATION MODELS FOR FMRI DATA MODELING

Recently, several foundational models for fMRI data modeling have emerged, leading fMRI research into a new paradigm. BrainLM (Ortega Caro et al., 2023) uses the MAE technique to segment brain signals by time intervals, embed them, and then reconstruct the signals to capture spatiotemporal representations. BrainMAE (Yang et al., 2024) follows a similar approach, representing each brain region individually over a time period and then combining the information from all brain regions for signal reconstruction. Dahan et al. (Dahan et al., 2024) proposed a surface-based MAE technique that focuses on reconstructing representations from short time intervals of signals. Armin et al. (Thomas et al., 2022) introduced a BERT and GPT-based approach for brain signal representation, leveraging prior knowledge of brain networks. These models utilize the MAE technique to reconstruct brain signals for capturing spatiotemporal representations, but they do not represent the whole-brain signals from the perspective of temporal signal prediction.

## 2.4 BRAIN FUNCTIONAL SIGNAL FORECASTING

Recent advancements in deep learning have significantly enhanced brain functional signal forecasting. Techniques such as convolutional neural networks (CNNs) (Xu et al., 2018), multi-scale CNNs (Yue et al., 2020), and hierarchical LSTM models (Liu et al., 2021) have been successfully applied to predict future fMRI volumes, effectively capturing spatial patterns, temporal dynamics, and hierarchical structures of brain signals. Graph neural networks (GNNs) have emerged as a promising approach for modeling the complex topological structure of brain networks; spatio-temporal GNNs (Wang et al., 2022) and dynamic GNNs (Zhong et al., 2023) incorporate both spatial and temporal dependencies, as well as the time-varying nature of brain connectivity, leading to more accurate predictions. Additionally, integrating prior knowledge and multi-modal information has further advanced the field: physiologically informed CNNs (Zhu et al., 2019) and anatomically informed GNNs (Chen et al., 2020) leverage domain expertise for more interpretable and biologically plausible outcomes, while multi-modal fusion frameworks (Sun et al., 2021) combine fMRI, EEG, and MEG signals to exploit complementary information from different imaging modalities. These developments not only enhance the predictive capabilities of deep learning models for brain signals but also contribute to a deeper understanding of neural mechanisms, thereby advancing the field of neuroscience and opening new avenues for clinical applications.

# 3 METHODOLOGY

## 3.1 FMRI DATA AND WHOLE BRAIN SIGNALS EMBEDDING

Functional magnetic resonance imaging (fMRI) signals are collected across multiple brain regions in a series of 3D volumes, at distinct time intervals known as repetition times (TRs). Each TR represents a snapshot of brain function at a specific moment, typically ranging from 1 to 3 seconds in length, depending on the scanning protocol used. The dynamic interactions between brain regions form what we refer to as brain networks—functional units responsible for various cognitive processes. Unlike prior methods (Thomas et al., 2022), which directly utilized brain networks derived using techniques like seed-based correlation or independent component analysis (ICA) (Maglanoc et al., 2020), our model utilizes raw whole-brain signals, thereby avoiding the risks associated with prior knowledge, and allows for a more flexible and accurate representation of the brain's natural connectivity structure. In our model, we use a parcellation of 1000 brain regions (Yan et al., 2023) to capture the fine-grained structure of brain activity which strikes a balance between the resolution of whole-brain data and the computational efficiency required for large-scale modeling (Glasser et al., 2016). To this end, first, the BrainSF takes as input a parcellated BOLD sequence $X \in \mathbb{R}^{T \times N}$, where $T$ represents the length of time points and $N$ denotes the brain regions (was set to 1000 in the subsequent experiments). Then, as shown in Figure 1, the brain signals at each time point are embedded into $\text{Emb}_{S_i} \in \mathbb{R}^{1 \times e}$ (e was set to 1024. In the figure, it is composed of colored blocks), which can be seen as a brain network learner. The brain signal representations at all time points are represented as $\text{Emb}_S \in \mathbb{R}^{T \times e}$.

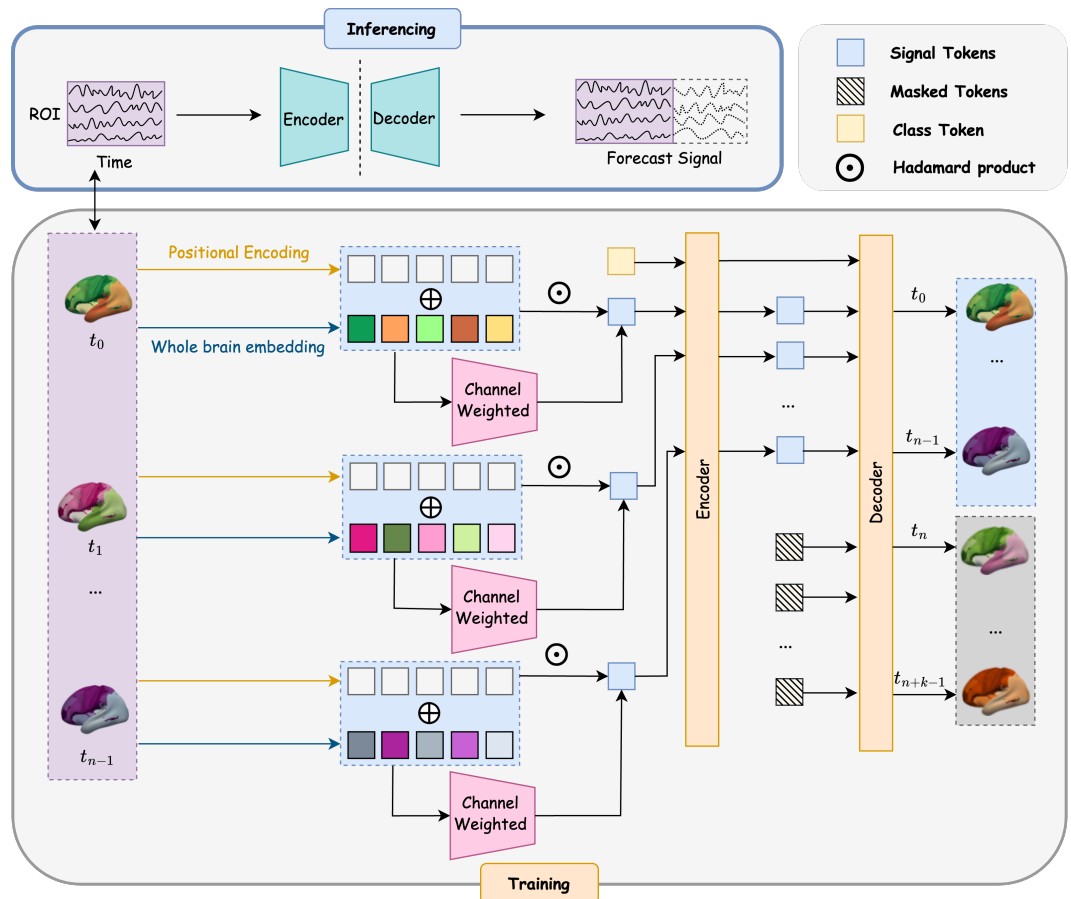

Figure 1: The framework and data flow of BrainSF. During the training phase, the data undergoes embedding, encoding, and decoding modules, and the final output is compared with the input to compute the loss. In the inference phase, the model parameters are frozen. During the downstream task phase, the parameters are either fully or partially frozen, and the CLS token along with signal embeddings are used for analysis.

## 3.2 ENCODER AND DECODER

A positional encoding is performed on every time point $t$ to obtain the position embedding $\text{Emb}_{T_i} \in \mathbb{R}^{1 \times e}$. Here, we use a learnable encoding strategy to fully capture the temporal relationships between the preceding and subsequent signals. The positional embedding and whole brain signal embedding for each time point are then summed to form the input data embedding $\text{Emb}_i$. This $\text{Emb}_i$ is passed to a channel weighting module, which calculates the weight of each representation dimension based on a learnable self-attention block. As mentioned earlier, learning the temporal dynamics and interactions of brain region signals is essential for representing brain functional signals. Here, we further compress and extract the previously $\text{Emb}_i$ by channel weighting to support the subsequent time series learning. Then, the obtained weights were directly multiplied by the previously obtained $\text{Emb}_i$ to produce the final embedding $\text{Emb}_{W_i}$ for the signal of $T_i$.

Our goal is to forecast the next $k$ time points based on signals from $n$ input time points. After obtaining embeddings for all input time points, we prepend a learnable class (cls) token to the sequence to capture latent features of the brain signals across all time points. These $n+1$ embeddings are then used as input to BrainSF's encoder, as shown in Equation 1. We set $k$ such that $\frac{k}{n+k} = 0.3$, resulting in $k = \frac{3}{7}n$. The encoder employs the standard Transformer architecture (Vaswani, 2017), consisting of 8 layers, each with 8 attention heads.

$$E = f(\text{cls}, \text{Emb}_{W_0}, \text{Emb}_{W_1}, \dots, \text{Emb}_{W_{n-1}}) \tag{1}$$

$$D = f(\text{cls}, \text{Emb}_{W_0}, \dots, \text{Emb}_{W_{n-1}}, \text{Masked}_n, \dots, \text{Masked}_{n+k-1}) \tag{2}$$

Transformers offer several key advantages over RNNs (Cho et al., 2014) when handling fMRI data and our experiments have validated. First, transformers rely on self-attention mechanisms, which enable them to capture long-range dependencies in the data more effectively than RNNs. This is particularly beneficial for fMRI data, where brain signals may exhibit relationships between distant time points. Additionally, the ability of self-attention mechanisms to model global interactions between different brain regions simultaneously can provide richer representations of functional connectivity. In contrast, RNNs primarily focus on local temporal dependencies, which may miss important cross-regional interactions. The flexibility in capturing non-sequential interactions and long-term dependencies makes transformers a powerful tool for analyzing complex, high-dimensional fMRI data.

The encoder's output is combined with $k$ masked tokens, re-applies positional encoding, places the cls token at the front, and then sends the $n+k+1$ tokens to the decoder as shown in Equation 2. The decoder also utilize the encoder architecture from the traditional transformer model (Vaswani, 2017) consist of 8 layers, with each layer featuring 8 attention heads. It is important to note that in methods like masked autoencoder (MAE) (He et al., 2022), the decoder is typically designed to be smaller than the encoder to efficiently learn the representation of the input information. However, in our case, we aim to fully integrate the information from the encoder with the signals to be predicted in order to enhance prediction performance. Therefore, we set the decoder's size to be the same as the encoder's. The selection of the number of attention heads and layers is based on a trade-off between model performance and computational resources. The current numbers are what our computational resources can handle, and we have found that as the model size increases, its performance also improves. We have demonstrated this in our ablation experiments.

Finally, the decoder's output, excluding the cls embedding, is linearly mapped to generate predictions for the $n + k$ time points. The first $n$ outputs correspond to the reconstruction of the input signals, while the last $k$ outputs correspond to the forecasted future signals. To balance the reconstruction and prediction tasks and to prevent overfitting, we define the loss function of BrainSF as the mean squared error (MSE), combining both the reconstruction loss of the input and the prediction loss for the future signals, as shown in Equations 3–5. In our experiments, we set the weighting parameter $\lambda = 0.75$.

$$\mathcal{L} = \lambda \mathcal{L}_{\text{input}} + (1 - \lambda) \mathcal{L}_{\text{forecast}} \tag{3}$$

$$\mathcal{L}_{\text{input}} = \frac{1}{n} \sum_{i=0}^{n-1} \left\| \hat{\boldsymbol{X}}_i - \boldsymbol{X}_i \right\|^2 \tag{4}$$

$$\mathcal{L}_{\text{forecast}} = \frac{1}{k} \sum_{i=n}^{n+k-1} \left\| \hat{\boldsymbol{X}}_i - \boldsymbol{X}_i \right\|^2 \tag{5}$$

### 3.3 IMPLEMENTATION DETAILS

During the pretraining phase, we randomly select 30-110 TRs of fMRI signals from the original sample each time, using the first 70% as the input signal and predict the subsequent 30%. We train all models using the ADAM optimizer with the following parameters: $\beta_1 = 0.9$, $\beta_2 = 0.999$, and $\epsilon = 1 \times 10^{-7}$. The training process consists of 15,000 epochs and involves a total of 109,978,984 trainable parameters. The training batch size is set to 256, and the learning rate is $5 \times 10^{-5}$, with a warm-up ratio of 0.001. The training phase was conducted on a single NVIDIA GeForce RTX 4090 GPU, utilizing 256 CPU threads and 90 GB of RAM, and took approximately 9 hours to complete.

### 3.4 VALIDATION OF THE PRETRAINED BRAINSF

Upon completing model training, we conducted a series of validation tasks to assess the performance of BrainSF. First, we evaluated the model's forecasting ability by predicting whole-brain signals on the testing dataset, using both the pretrained and fine-tuned versions of BrainSF. These results were then benchmarked against other existing methods applied to the same dataset to ensure a comprehensive comparison. Subsequently, we assessed the model's generalization capability in a zero-shot learning setting, testing its performance on out-of-domain data. In addition, we investigated how the model performed when predicting signals of varying temporal lengths, providing insight into its robustness over different time scales. Finally, the latent embeddings generated by the model were utilized for mental state decoding on unseen HCP-task data, further demonstrating the model's versatility in neuroimaging tasks.

## 4 EXPERIMENTS

### 4.1 DATASETS AND PREPROCESSING

#### 4.1.1 DATASETS FOR MODEL TRAINING AND VALIDATION

**HCP Resting-State Data (HCP-rs).** We used resting-state fMRI (rs-fMRI) data from the Human Connectome Project (HCP) 1200-subject release (Van Essen et al., 2013b). A total of 1,500 scans from 375 subjects who completed all four runs (scanned up to four times, twice on one day and twice on a second day) were selected for training and validation. The data were acquired with a temporal resolution (TR) of 0.72 s and a duration of 1,200 frames per run (14.4 minutes), resulting in a total of 1,800,000 TRs.

**HCP Movie-Watching Data (HCP-movie).** We included 709 scans from the HCP 7T release (Griffanti et al., 2014), which collected fMRI data during movie-watching sessions. These data have a TR of 1 s and each run has over 900 frames (15 minutes), amounting to a total of 647,902 TRs.

**Narratives Data.** The "Narratives" collection (Nastase et al., 2021a) aggregates auditory story-listening fMRI datasets with a TR of 1.5 s. It provides approximately 4.6 hours of unique auditory stimuli (11,149 TRs). Combined across all subjects, the dataset encompasses roughly 6.4 days of fMRI data, totaling 369,496 TRs.

#### 4.1.2 DATASETS FOR ZERO-SHOT LEARNING AND DOWNSTREAM TASKS

**HCP Task-Based Data (HCP-task).** A subset of task-based fMRI data from the HCP 1200-subject release was used, consisting of 350 subjects who completed seven tasks: emotion, gambling, language, motor, relational, social, and working memory (WM). A total of 2,451 scans were collected, with a TR of 0.72 s. Each run contains between 176 and 405 TRs, resulting in approximately 670,810 TRs in total. For detailed task names, subtask names, and corresponding times, please refer to the appendix.

**CHCP Resting-State Data (CHCP-rs).** We selected a subset of the resting-state fMRI dataset from the Chinese Human Connectome Project (CHCP) (Ge et al., 2023) as a zero-shot testing dataset. This subset consists of 70 subjects, each with data acquired at a TR of 0.71 s and a duration of 1,200 frames per run, totaling 84,000 TRs.

**CHCP Task-Based Data (CHCP-task).** Similar to the HCP task data, a subset of task-based fMRI data from CHCP was used, consisting of 102 subjects who completed seven tasks. A total of 2,451 scans were collected, with a TR of 0.72 s. Each run contains between 288 and 411 TRs, resulting in approximately 241,682 TRs in total.

#### 4.1.3 DATASET PREPROCESSING

All fMRI datasets underwent the HCP's generic fMRI volume minimal preprocessing pipeline and were registered to the MNI152 common space (Glasser et al., 2013). This was followed by filtering and normalization procedures. We extracted 1,000 cortical regions of interest (ROIs) from

the homotopic atlas (Yan et al., 2023), resulting in data with dimensions of $1,000 \times$ TRs for each sample.

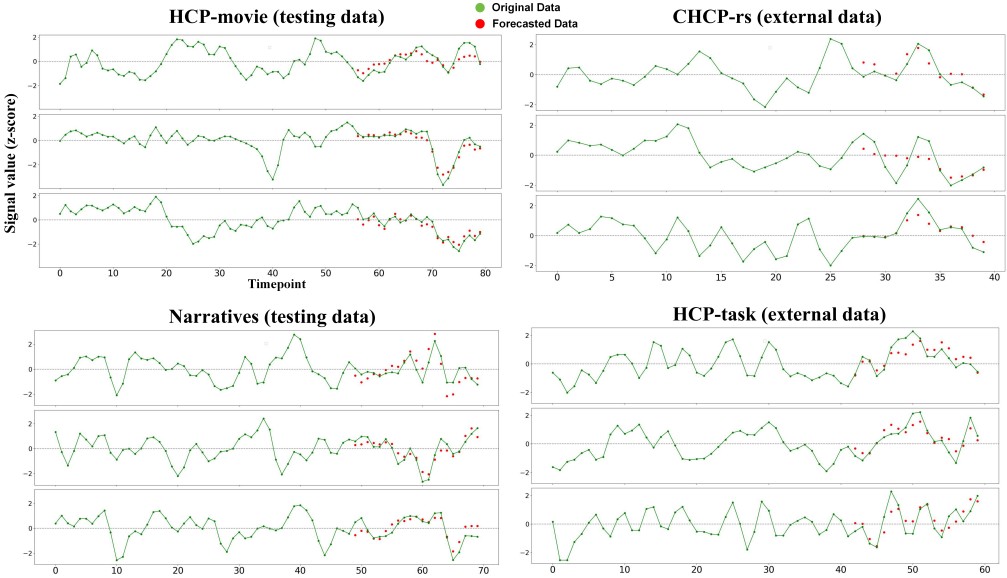

Figure 2: Validation of forecasting results of four test sample, here only three brain regions were showed. The green points are ground truth, and the red points are forecasted values.

## 4.2 TESTING RESULTS

We used 80% of the HCP resting-state fMRI data and natural stimulation fMRI data for training and then tested on the remaining data. Figure 2 illustrates the signal prediction results of the pre-trained BrainSF on various unseen test set data and out-of-domain datasets. The model's performance is demonstrated using different input and output lengths. For clarity, only three brain regions are shown for each sample. Since there is no foundation model for brain time series signal prediction, we use MLP, RNN, and LSTM as comparison baseline methods. We trained these methods on the same training dataset. For the convenience of comparison, these methods take the same length of signal as input and forecast the same length of future signals. As shown in Table 1, all models use signals at 35 time points as input and output signals at 15 time points as forecasted values. We compared them across three different types of datasets, presenting the results using metrics such as $R^2$ (Coefficient of Determination), $R$ (Pearson Correlation Coefficient), and MSE over whole-brain ROIs. We evaluated the model by directly applying the pre-trained BrainSF to the test set, as well as by fine-tuning the model with fixed input and output lengths before testing. It can be observed that the performance of the pre-trained BrainSF surpasses other models, and after fine-tuning, its performance significantly outperforms the others.

Zero-shot learning capability is a crucial attribute of a foundational model, particularly for brain functional signal prediction. A robust foundational model should learn generalized representations of how signals evolve across different brain regions over time. As previously discussed, we conducted zero-shot validation of BrainSF's prediction performance on three out-of-domain datasets: HCP-task, CHCP-rs, and CHCP-task. Additionally, we compared these results with BrainSF's forecasting performance on unseen HCP-rs test data. As shown in Table 2, the model's performance on out-of-domain data closely mirrors its performance on the test set. Notably, in the case of resting-state data, the zero-shot performance even exceeds that on the test set.

To further evaluate BrainSF's robustness across varying prediction lengths and to explore the upper limits of signal prediction, we compared its performance on unseen HCP-rs test data with prediction lengths ranging from 12 to 30 time points. As demonstrated in Table 3, the model achieves its best performance with a prediction length of 12. However, as the prediction length increases, performance gradually declines, with a sharp drop at a length of 30. This degradation can be attributed

Table 1: Comparison of models across HCP-rs, HCP-movie and Narratives datasets. $R^2$ (Coefficient of Determination), $R$ (Pearson Correlation Coefficient), and MSE across whole-brain ROIs are reported. The best and second results are highlighted. † indicates the corresponding model was fine-tuned.

| Methods | HCP-rs | | | HCP-movie | | | Narratives | | |
|---------|--------|--------|--------|-----------|--------|--------|------------|--------|--------|
| | $R^2$ | R | MSE | $R^2$ | R | MSE | $R^2$ | R | MSE |
| RNN | 0.091 | 0.306 | 0.946 | 0.112 | 0.339 | 0.920 | 0.006 | 0.098 | 0.990 |
| LSTM | 0.191 | 0.437 | 0.846 | 0.202 | 0.450 | 0.793 | 0.051 | 0.225 | 0.926 |
| MLP | 0.447 | 0.669 | 0.558 | 0.396 | 0.629 | 0.615 | 0.211 | 0.461 | 0.784 |
| BrainSF | 0.436 | 0.658 | 0.527 | 0.609 | 0.780 | 0.363 | 0.605 | 0.771 | 0.381 |
| BrainSF† | 0.654 | 0.808 | 0.324 | 0.782 | 0.884 | 0.202 | 0.738 | 0.857 | 0.253 |

to two factors: first, the high complexity of brain spatiotemporal signals, which makes longer-term predictions more challenging, particularly when using shorter input signals that may fail to capture intricate brain state representations (Foster & Scheinost, 2024). Second, the model may be limited by the positional encoding constraints of the pretrained architecture. This limitation could potentially be mitigated by increasing the input signal length during training, thereby improving the model's ability to handle longer prediction horizons.

Table 2: Zero-shot learning forecasting results on three external datasets. Coefficient of Determination ($R^2$), Pearson Correlation Coefficient ($R$), and Mean Squared Error (MSE) across whole-brain ROIs metrics are reported.

| Dataset | $R^2$ | R | MSE |
|---------|-------|-------|-------|
| HCP-rs (unseen testing) | 0.436 | 0.658 | 0.527 |
| HCP-task (external) | 0.438 | 0.661 | 0.523 |
| CHCP-rs (external) | 0.452 | 0.670 | 0.522 |
| CHCP-task (external) | 0.532 | 0.728 | 0.459 |

Table 3: Comparison of different forecasted length on HCP resting-state unseen testing dataset. Coefficient of Determination ($R^2$), Pearson Correlation Coefficient ($R$), and Mean Squared Error (MSE) across whole-brain ROIs metrics are reported.

| Forecasted Length | $R^2$ | R | MSE |
|-------------------|-------|-------|-------|
| 12 | 0.496 | 0.703 | 0.470 |
| 18 | 0.463 | 0.678 | 0.533 |
| 20 | 0.446 | 0.665 | 0.548 |
| 25 | 0.441 | 0.660 | 0.551 |
| 30 | 0.236 | 0.488 | 0.759 |

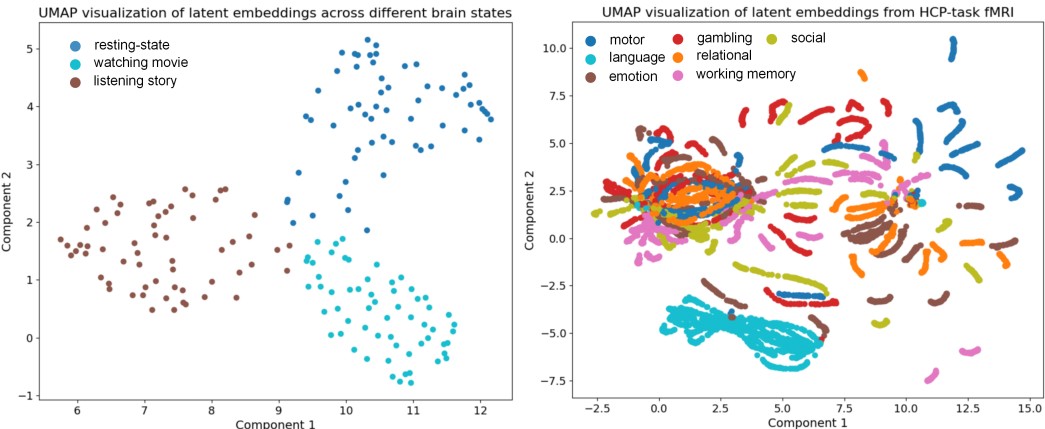

Figure 3: The visualization of latent embedding. The left panel shows the UMAP visualization of latent embeddings across three datasets, and the right shows the atent embedding of seven brain states from HCP-task dataset.

## 4.3 INTERPRETATION AND DOWNSTREAM TASK

To explore the reliability of the model's representation of fMRI data and its performance on downstream tasks, we visualized the latent embeddings of various types of data and then performed decoding on a specific fMRI task-based mental state. The main goal of this work is to forecast brain function signals, but we also explored the representations learned by the model. We only trained the BrainSF on natural stimulation and resting state data, without any task state data. We hope to verify that the BrainSF learns a general and rich spatiotemporal representation from natural stimulation and resting state data that can also be used for task state fMRI data.

### 4.3.1 LATENT EMBEDDING OF TIME-SERIES SIGNALS

As illustrated in the left panel of Figure 3, we selected one sample from each of the three test datasets—HCP-resting, HCP-movie, and Narratives—each consisting of 60 time points. These samples were input into the pre-trained BrainSF model to extract the final encoder output, which serves as the latent embedding. Each time point's embedding is 1024-dimensional, resulting in a 60x1024 matrix for each sample. To visualize the relationships among the time points, we applied UMAP for dimensionality reduction, projecting the embeddings of all 180 time points into a two-dimensional space. The visualization clearly shows that data from the three distinct brain states cluster into separate regions, with embeddings from the same brain state grouped closely together. To further evaluate the model's performance on out-of-domain data, we applied the same embedding procedure to HCP-task data. We randomly selected task data from 20 subjects across 7 major task categories, with each sample representing signals recorded during the execution of a specific task, and the number of time points varied across tasks. As shown in the right panel of Figure 3, time point embeddings within each task form well-defined clusters, often aligning along a linear structure, and the embeddings from different task types are clearly separated. This indicates the model's strong capability to effectively capture and represent temporal sequences from diverse and unseen data sources.

### 4.3.2 MENTAL STATE DECODING

To further evaluate BrainSF's decoding capability on fine-grained, task-based data, we performed a classification task on 19 distinct mental states from the HCP-task dataset. Detailed descriptions of these 19 mental states, including their 7 major categories, subcategories, and corresponding signal acquisition times, are provided in Appendix Table 6. In addition to task data, we also included resting-state data, forming a total of 20 mental state classes for classification, following a similar approach to previous studies (Thomas et al., 2022). The classification process involved two approaches. First, we employed a linear probing technique (He et al., 2022), where the parameters of the pre-trained BrainSF model were frozen. The CLS token for each sample was extracted from the model and passed through a learnable linear mapping layer to classify the sample into its respective mental state. In the second approach, we allowed the CLS token to be trainable, optimizing it alongside the classification task.

Since previous methods were specifically trained on HCP-task data or other task-based datasets, direct comparisons with our approach were not feasible. As a result, we used MLP as the baseline for comparison. Table 4 presents the classification accuracy and F1 scores for both approaches. The results show that the linear probing method, where BrainSF's parameters are frozen, significantly outperforms the MLP baseline. Additionally, when the CLS token is fine-tuned, the classification performance improves even further, demonstrating the model's strong capability in decoding mental states.

Table 4: Results for transient mental state decoding. The Accuracy (Acc.) and Macro F1-score (F1) metrics are reported † indicates the corresponding model was fine-tuned.

| Model | Acc. (%) | F1 |
|---|---|---|
| MLP | 51.9 (±0.79) | 55.6 |
| BrainSF | 85.6 (±0.84) | 82.6 |
| BrainSF † | 87.5 (±0.14) | 83.1 |

## 4.4 ABLATION STUDY

We conducted ablation experiments on model size and channel weight modules for two tasks: signal forecasting and mental state decoding. The results confirmed that as the model size increases, its performance improves. Additionally, we found that adding the channel weight module significantly boosts the model's performance across various tasks. For detailed results are shown in Table 5.

Table 5: Ablation studies on model size and Channel Weighted module for signal forecasting and transient mental state decoding tasks. Coefficient of Determination ($R^2$), Pearson Correlation Coefficient ($R$), Mean Squared Error (MSE) across whole-brain ROIs, Accuracy (Acc.) and Macro F1-score (F1) metrics are reported.

| Model Configuration | Signal Forecasting (HCP-rs) | | | Mental State Decoding (HCP-task) | |
|---|---|---|---|---|---|
| | $R^2$ | R | MSE | Acc.(%) | F1 |
| 60M Parameters | 0.375 | 0.592 | 0.612 | 79.6 (±0.41) | 75.3 |
| 80M Parameters | 0.415 | 0.641 | 0.543 | 83.1 (±0.22) | 80.7 |
| 110M Parameters | 0.436 | 0.658 | 0.527 | 87.5 (±.014) | 83.1 |
| w. channel weighted | 0.436 | 0.658 | 0.527 | 87.5 (±0.14) | 83.1 |
| w/o. channel weighted | 0.402 | 0.638 | 0.534 | 82.2 (±0.64) | 78.5 |

The ablation experiments provide important insights into model performance. First, the scaling laws of transformers also apply to our model, suggesting room for improvement, particularly with the use of larger datasets and more parameters. This indicates that increasing model capacity could lead to better generalization and accuracy. Second, the self-attention mechanism proves effective in capturing the intricate interactions between brain region signals. By leveraging relative spatial representations of brain regions before conducting temporal modeling, we strike a balance between capturing spatial and temporal dynamics.

Future research could focus on expanding the model size to fully harness the potential of transformers for fMRI time-series data representation. Additionally, optimizing the self-attention mechanism could further enhance the modeling of brain region interactions, leading to more refined signal embeddings that better prepare the model for subsequent temporal prediction tasks. This combination of scaling and attention optimization may help push the boundaries of performance in brain signal prediction and mental state decoding tasks.

## 5 CONCLUSION

In this work, we propose a foundational model for forecasting whole brain functional signals with multi-scale input and output trained on large-scale fMRI data. The model can fully exploit the temporal information of brain signals by separately learning representations of the fMRI signals at each time point, then representing the input tokens in the encoder module and merging the forecasted tokens in the decoder module. The signal forecasting results on the test data and external data demonstrate the model's strong representation performance and generalization capability. Surprisingly, even without training on any task-based fMRI data, the model is still able to produce strong temporal representations for such data, particularly demonstrating notable performance in mental state decoding. It can be found that naturalistic stimuli and resting-state fMRI data may contain sufficient representations of brain functional networks information, which can be easily transferred to various task-state data.

Our model has the potential for further exploration and significant breakthroughs in brain cognition research and the study of brain diseases through its refined representation of brain state information. Currently, the decoding stage only includes positional information, but in the future, various prior knowledge can be integrated into the decoding stage as needed to enhance the model's interpretability and causal representation capabilities.

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

# A APPENDIX

## A.1 ADDITIONAL ANALYSIS RESULTS

We offer a concise summary of the mental states featured in the downstream datasets below. For additional information on the experimental protocols, please refer to the original publications (Van Essen et al., 2013b).

Table 6: Mental states and corresponding durations for different tasks.

| Task | Sub task | Duration (seconds) |
|---|---|---|
| Working Memory | body | 27.5 |
| | faces | 27.5 |
| | places | 27.5 |
| | tools | 27.5 |
| Gambling | win | 28.0 |
| | loss | 28.0 |
| Motor | left finger | 12.0 |
| | right finger | 12.0 |
| | left toe | 12.0 |
| | right toe | 12.0 |
| | tongue | 12.0 |
| Language | story | 25.9 |
| | math | 16.0 |
| Social | interaction | 23.0 |
| | no interaction | 23.0 |
| Relational | relational | 16.0 |
| | matching | 16.0 |
| Emotion | fear | 18.0 |
| | neutral | 18.0 |

## A.2 FMRI, BRAIN NETWORK AND PARCELLATION

Functional magnetic resonance imaging (fMRI) is a non-invasive neuroimaging technique that measures brain activity by detecting changes in blood oxygenation levels, which correlate with neural activity over time. fMRI data is collected in a series of time points, known as "time repetitions" (TRs), where each TR represents a snapshot of the brain's activity at a specific moment . This temporal resolution enables researchers to track dynamic changes in brain activity across different regions. A key application of fMRI is the study of brain functional networks, which are groups of brain regions that show synchronized activity patterns over time. These networks, including well-known ones like the default mode network and the frontoparietal network, are critical for various cognitive and behavioral functions. Analyzing fMRI signals at different TRs provides insights into the temporal interactions between these networks, helping researchers map the brain's functional organization and connectivity .

Resting state fMRI data from 1489 subjects were registered using surface-based alignment. A gradient weighted markov random field approach was employed to identify cortical parcels ranging from 100 to 1000 parcels, the 1000 parcels was adopted in this work, more details are available in (Yan et al., 2023).

## A.3 LIMITATIONS

Despite the significant achievements of this work, several limitations remain. First, while the model performs well on the HCP tasks, its ability to generalize to a broader range of cognitive tasks and states requires further validation. Additionally, the current temporal resolution (0.72-1.5 seconds)

may not be adequate for capturing fast, millisecond-level neural dynamics. Regarding spatial resolution, although the division into 1000 regions balances spatial specificity and computational efficiency, it may miss finer spatial patterns. Another limitation is the model's interpretability, which still needs improvement. Further research is required to map the learned representations to known functional brain networks and cognitive processes.

Moreover, the training data primarily consist of young, healthy adults from the HCP dataset, so the model's performance in clinical populations, across different age groups, or in more diverse demographic samples remains uncertain. While BrainSF can predict future brain states, it does not provide insights into the causal relationships between different brain regions over time. The model's substantial computational demands may also limit its accessibility and applicability in certain research or clinical settings. Furthermore, due to computational constraints, we did not perform an exhaustive search of model architectures and hyperparameters, which could have impacted the model's optimal performance.

These limitations do not diminish the promising potential of this approach for modeling whole-brain functional dynamics. Future work will focus on addressing these challenges to further enhance the foundational model's utility in both neuroscience research and clinical applications.

