# OpenReview forum: "BrainSF: A Foundation Model for Whole Brain Functional Signals Forecasting"
_ICLR.cc/2025/Conference — ICLR 2025 Conference Withdrawn Submission_

### Official Review · Reviewer_XT3V · 2024-11-03

**Soundness:** 2
**Presentation:** 3
**Contribution:** 2
**Rating:** 5
**Confidence:** 4

**Summary:**

The paper introduces a self-supervised model designed to forecast whole-brain fMRI signal sequences, addressing a gap in current methods that primarily characterize observed brain signals without predicting future signals. The model embeds momentary whole-brain fMRI signals into vector representations and is trained on a large-scale dataset that includes resting-state and naturalistic stimuli conditions. Experimental results demonstrate its effectiveness in zero-shot forecasting of future signals on unseen data and strong performance in downstream tasks such as task-based functional state decoding. This work claims to be the first approach of its kind at such a scale, with potential implications for theoretical research on brain signal dynamics and clinical applications in diagnosing and treating brain disorders.

**Strengths:**

ChatGPT
The paper presents a novel self-supervised model capable of forecasting whole-brain fMRI signal sequences, addressing a critical gap in existing methods that typically focus on characterizing observed signals. Its strong performance in zero-shot forecasting and task-based functional state decoding highlights the model's potential to advance research on brain dynamics and offer valuable insights for clinical applications.

**Weaknesses:**

1. Lack of Discussion on Computational Time: The paper does not address the computational time required for processing large-scale fMRI datasets, which is a critical factor given the substantial number of time points involved. This omission makes it difficult to evaluate the practicality of the proposed model in real-world applications.

2. Limited Utility of Time Point Reconstruction: While the model focuses on reconstructing future time points, this approach alone may not provide sufficient insights into brain dynamics. The authors should incorporate downstream tasks to validate the model's performance more effectively, such as analyzing functional connectivity in resting-state fMRI or examining activation maps in task-based fMRI studies.

3. Insufficient Validation through Diverse Applications: The experimental results primarily demonstrate zero-shot forecasting, but the lack of validation through additional applications, such as functional connectivity analyses, limits the understanding of the model's practical utility in broader neuroscience research contexts. Incorporating such analyses would strengthen the case for the model's effectiveness and relevance.

**Questions:**

1. Limited Practical Utility: While the self-supervised model addresses the forecasting of whole-brain fMRI signals, it lacks practical applications in validating its effectiveness. The paper does not incorporate essential downstream tasks, such as functional connectivity analysis or activation mapping, which are crucial for demonstrating the model's relevance in real-world neuroscience research.

2. Omission of Computational Time Considerations: The paper fails to address the computational time required for processing large-scale fMRI datasets. Given the extensive number of time points involved, this oversight raises concerns about the feasibility and efficiency of implementing the proposed model in practical scenarios.

3. Inadequate Justification of Contributions: The model's contributions are not sufficiently compelling to warrant acceptance, as the focus on reconstructing future time points alone does not provide enough insight into brain dynamics. Without stronger experimental validation and broader applicability, the significance of the proposed approach remains unclear.

---

### Official Review · Reviewer_iw6P · 2024-11-04

**Soundness:** 2
**Presentation:** 3
**Contribution:** 2
**Rating:** 3
**Confidence:** 4

**Summary:**

This study presents an approach to forecasting whole-brain fMRI signal sequences by developping a self-supervised model that converts momentary whole-brain fMRI signals into vector representations, enabling continuous prediction of future brain activity. Trained on a large fMRI dataset that includes both resting-state and naturalistic stimuli conditions, the model demonstrated the capability in zero-shot forecasting of unseen data and was effective in functional state decoding tasks.

**Strengths:**

- The motivation, objectives, and methodology of the paper are clearly articulated and well-organized.
- The research addresses an important and relevant problem within the field.
- The pretraining approach effectively utilizes a large-scale fMRI dataset.
- It is intriguing to observe that the model, trained on both resting-state and naturalistic datasets, demonstrates generalizability to task-phase data.

**Weaknesses:**

- A primary concern is the lack of baseline comparisons in this study. Although the authors claim that this is the only work addressing the fMRI forecasting problem, prior fMRI foundation models, such as BrainLM (Ortega Caro et al., 2023), which employs masked-prediction training, can also perform forecasting by placing the mask over the prediction segment. Additionally, the forecasting horizon in this study (up to 30 TR) aligns with the sequence length used in BrainLM (200 TR, if recalled correctly). The authors' rationale for not including a comparison with BrainLM is unclear.

- This absence of baselines raises a second concern: the model architecture used in this paper appears somewhat antiquated techniques. Without comparisons to other models, there is no evidence that the current methodology is more effective or novel for fMRI forecasting compared to contemporary time-series forecasting architectures. Beyond the MAE architecture previously applied to fMRI data, numerous advanced time-series forecasting models could be potential baselines that could be evaluated on fMRI data, such as those available in the “Time-Series Library” repository (https://github.com/thuml/Time-Series-Library/tree/main).

- The study does not address the influence of fMRI auto-correlation, a significant factor in fMRI forecasting. Typically, fMRI signals exhibit auto-correlation across a time lag of approximately -10 to 10 seconds, potentially making short-term forecasting within this range easier. In this study, the forecasting horizon for Table 1 is set to 15 TR, which largely falls within this auto-correlation window. While this does not imply an issue with the methodology itself, it would be valuable to examine prediction performance when introducing a gap between the input and forecasting time windows to reduce the effects of auto-correlation. At a minimum, the authors should discuss this factor, as it plays an important role in fMRI forecasting task.

**Questions:**

- The authors claim an innovative approach in modeling "whole-brain" activity. In this context, I anticipated predictions at the voxel level (providing high spatial resolution) or, at a minimum, coverage of both cortical and subcortical regions. However, this study continues to utilize cortical parcellations to convert the fMRI 4D volume into ROI-by-time time series, similar to existing studies, though with an increased number of parcellations. Since prior studies also covered the entire cortical region, this terminology appears somewhat misleading. Could the authors further clarify the specific distinctions in preprocessing or model design that differentiate this study as "whole-brain," and explain why this study is considered "whole-brain" while others are not?
- For line 214, and line 265, is there any rationale behind the choice of 0.3 for the prediction portion?
- For Figure 2, could the authors specify which regions are shown as examples? Adding clarifications in either the text or figure captions would be helpful.

---

### Official Review · Reviewer_WiVW · 2024-11-04

**Soundness:** 2
**Presentation:** 3
**Contribution:** 2
**Rating:** 3
**Confidence:** 4

**Summary:**

This paper presents a novel self-supervised model for forecasting whole-brain fMRI signals, addressing a gap in predicting continuous future brain activity. Trained on a large fMRI dataset, the model embeds brain signals into vector representations to forecast future states, showing good performance in zero-shot prediction on new data. It also excels in decoding functional states under task-based conditions.

**Strengths:**

1 - The model’s use of a spatiotemporal encoding architecture to capture both spatial and temporal information is a strong feature.

2 - The authors conduct extensive ablation studies that validate the model architecture’s contributions and highlight the performance impact of each component.

**Weaknesses:**

1 - The paper lacks clear organization, particularly in the Methods section, which requires a more detailed description of the model architecture to enable thorough evaluation.

2 - Given the brain's dynamic complexity and inherent chaotic behavior, foundational models should include mechanisms for capturing uncertainty in time-series data, such as using variational encoder-decoder architectures for sequence-to-sequence modeling or incorporating partial differential equation (PDE) terms in the loss function for physics-informed neural networks (PINNs). The proposed approach, however, does not address this crucial aspect.

3 - The abstract claims that the authors propose a self-supervised model to embed whole-brain fMRI signals into vector representations for continuous future prediction. However, the Methods section indicates the use of 1,000 parcels, which suggests a misalignment between the stated objective and actual implementation.

4 - The paper lacks a dedicated interpretability section.

5 - The reported generalization metrics ($R^2 , R, MSE$) require contextualization; how do we know they are good?

6 - The computational complexity of the model is significant, as highlighted in Section 3.3.

**Questions:**

1 - How many learnable parameters does the model contain? My primary concern is that the network architecture appears relatively modest in size for a foundational model, which could restrict its representational capacity.

2 - In Section 3.1, it is stated that "BrainSF takes as input a parcellated BOLD sequence $X \in \mathbb{R}^{T \times N}$". If $T$ represents the number of time points and $N$ denotes the number of parcels, what is the voxel count within each parcel?

3 - Could you provide additional details regarding the model architecture to enhance clarity and understanding?

4 - The proposed method mentions the utilization of 1,000 parcels. What is the rationale behind employing parcels instead of using the entire 4D tensor as input to the model?

---

### Note · Authors · 2024-11-22

I have read and agree with the venue's withdrawal policy on behalf of myself and my co-authors.